# RETHINKING ALIGNMENT IN CROSS-LINGUAL KNOWLEDGE TRANSFER

## ABSTRACT

Despite LLMs' advanced performance in multilingual tasks, they usually have performance gap on the same task in dominant languages (e.g., English) and non-dominant languages (e.g., Turkish). In this paper, we analyze LLMs' capacity to transfer the task knowledge learned in the dominant language to the non-dominant language. We first formulate the cross-lingual transfer problem into a gradient alignment problem and then connect it to the representation alignment problem. We show that pre-trained LLMs with decent representation alignment ability can easily transfer knowledge from the dominant language by simple fine-tuning, while others need carefully designed training strategies. For the latter, we propose a cross-lingual in-context prompt tuning (CL-ICP) model to enhance gradient alignment, which utilizes in-context attentions to generalize to unseen data. In addition, we apply a representation shift to enhance representation alignment between demonstration and target samples. Experiments show that CL-ICP improves cross-lingual transfer in both high and low resource scenarios. The code is available in `https://anonymous.4open.science/r/Cross-Lingual-Alignment-94C2`.

## 1 INTRODUCTION

State of the art LLMs are pre-trained to have strong capacities in solving multilingual downstream tasks (Shi et al., 2023; Zhao et al., 2025a). However, their performance varies across different languages even for the same task (Huang et al., 2023). Specifically, models tend to perform better in dominant languages like English than in non-dominant languages like Turkish (Asai et al., 2024; Gurgurov et al., 2024a). Assuming the major knowledge for solving a task is language agnostic, models performing well in the dominant language should have the knowledge to solve the task and are supposed to also perform well in non-dominant languages. In this paper, we study the cross-lingual transfer problem which utilizing the knowledge learned from a dominant language to enhance the model's task performance in a non-dominant language.

Previous works have identified that models' alignment between languages is the key to cross-lingual transfer (Stap et al., 2023; Tanwar et al., 2023a; Qin et al., 2024; Zeng et al., 2025). The multi-lingual alignment includes: (1) language-level alignment, which utilizes LLMs' ability to encode the knowledge from different languages to the dominant language's space (Wang et al., 2024; Zhao et al., 2024b; Wendler et al., 2024; Schut et al., 2025); (2) representation-level alignment, where models are trained to connect data representations from different languages in supervised and unsupervised manners (Bornea et al., 2021; Li et al., 2024; Zhang et al., 2025a). However, what kind of alignment is necessary for knowledge transfer remains underexplored (Hämmerl et al., 2024). Such lack of understanding presents a challenge in optimizing cross-lingual transfer strategies. For example, in post-training, some works claim that simple fine-tuning on the dominant language is already strong for cross-lingual transfer (Chirkova & Nikoulina, 2024); while other works emphasize the need for additional alignment between language representations (Li et al., 2024; Zhang et al., 2025a).

To better understand the effect of alignment in cross-lingual transfer, we first formulate cross-lingual transfer as a gradient alignment problem that aims to maximize the inner product between gradients on the dominant and non-dominant languages' data during training (Riemer et al., 2019). Therefore, the decrease of loss on the dominant language's data will also lead to a loss decrease on the non-dominant language's data and cause knowledge transfer. Then we show that gradient alignment

depends on the alignment between representations of dominant and non-dominant languages' data. For pre-trained LLMs with decent representation alignment ability, fine-tuning on the dominant language may directly transfer the knowledge to the non-dominant language.

In the case that needs further tuning with the non-dominant language's data, we show that simple fine-tuning and multi-task learning tend to maximize gradient alignment within and across languages and therefore have strong performance with sufficient training data. However, data in non-dominant languages are usually low-resourced (Huang et al., 2023). In the low-resource scenario, existing models which enforce alignment between accessible data may not be generalized enough. To address this problem, we propose a cross lingual in-context prompt tuning (CL-ICP) model which utilizes the target sample's attention on different demonstration samples to infer its relation to the unseen data. When training with in-context samples, it enforces gradient alignment based on the contribution of each dominant language's data (demonstration) to the prediction of non-dominant language's data (target). To improve the relevance between demonstration and target samples for better transfer, we shift the target sample's representation to make it closer to the demonstration's representation. Experiments show that CL-ICP outperforms existing models in both high and low resource scenarios.

In summary, our work makes the following contributions:

- We provide a new and unified perspective for studying the alignment problem in cross-lingual transfer. Specifically, we formulate the problem with a gradient alignment objective; analyze the connection between gradient and representation alignment; and show the alignment effects in existing models.
- We show that adding in-context samples may further improve the gradient alignment during training; and propose a CL-ICP model with representation shift to improve both gradient and representation alignment.
- Experiments in different cross-lingual transfer scenarios show the effectiveness of our CL-ICP model, which further supports our alignment claims.

## 2 RELATED WORK

**Cross-lingual Transfer**   Despite strong multilingual capacity (Shi et al., 2023; Zhao et al., 2025a), current LLMs usually have limited proficiency in low-resource languages (Huang et al., 2023; Gurgurov et al., 2024a). This calls for cross-lingual transfer models that leverage knowledge from high-resource (dominant) languages to improve performance in low-resource (non-donimant) languages (Ruder et al., 2019). Existing works improve cross-lingual transfer from the pre-training to post-training stages. Conneau & Lample (2019); Ouyang et al. (2021) use translation based pre-training to enhance LLMs' multilingual transfer ability. Gupta et al. (2023); Zhao et al. (2024a); Fujii et al. (2024) continually pre-train the LLMs to better understand non-dominant languages. Pan et al. (2020); Wang et al. (2022); Gurgurov et al. (2024b); Cassano et al. (2024) use adaptation techniques to transfer pre-trained knowledge to non-dominant languages' tasks.

Recently, In-Context Learning (ICL) has emerged as a powerful paradigm for LLM post training (Brown et al., 2020). Previous works have also explored the benefit of ICL in multilingual task learning (Li et al., 2024), but only few of them explored the transfer ability of using cross-lingual samples in ICL (Tanwar et al., 2023b; Zhang et al., 2024). Even for the above works, they aimed to improve ICL in the multilingual settings without training, and thus not exploring the cross-lingual knowledge transfer ability of in-context training compared to fine-tuning or multi-task learning. Our work analyzes the benefits of training with in-context samples in cross-lingual transfer, showing that it may achieve better alignment to unseen data and thus benefit in the low-resource scenario.

**Cross-Lingual Alignment**   Alignment has been recognized as a crucial factor in effective cross-lingual transfer (Stap et al., 2023; Tanwar et al., 2023a; Qin et al., 2024; Zeng et al., 2025). Previous works show that LLMs tend to map different languages' representations to one language's representation space (Wang et al., 2024; Zhao et al., 2024b; Wendler et al., 2024; Schut et al., 2025), which indicates their language alignment ability before post-training. In post-training, existing works add extra representation alignment to better align cross-lingual representations via translation-based objectives (Bornea et al., 2021), contrastive learning (Li et al., 2024) and representation shift (Zhang et al., 2025a; Sundar et al., 2025). However, to what extent the extra alignment is necessary in post-

training remains unexplored (Hämmerl et al., 2024). In this work, we first analyze the alignment in the gradient perspective and connect it to the representation alignment. Our analysis bridges existing training strategies and representation alignment methods, indicating the cases when representation alignment can compensate existing learning strategies for better cross-lingual transfer.

# 3 PROBLEM STATEMENT

In this paper, we study the cross-lingual transfer problem which utilizes the task knowledge learned in the dominant language to enhance the same task's performance in the non-dominant language.

## 3.1 CROSS-LINGUAL TRANSFER SETTINGS

We consider two main cross-lingual transfer scenarios: (1) **Dominant language direct transfer** where the model learns the task in the dominant language and then is directly applied to the non-dominant language; (2) **Mixed language transfer** where the model learns the task in both the dominant and non-dominant languages to improve performance in the non-dominant language.

Based on different data accessibility in the above scenarios, we divide the data in cross-lingual transfer into the following categories:

- The accessible data $\mathcal{D}_s$ in the dominant language $s$;
- The accessible data $\mathcal{D}_t$ in the non-dominant language $t$;
- The inaccessible data $\mathcal{D}_{unseen}$, which is a minimal complementary set that enables the LLM to sufficiently learn the task in language $t$ when added to the accessible data.

In the dominant language direct transfer only $\mathcal{D}_s$ is used in training, while in the mixed language transfer both $\mathcal{D}_s$ and $\mathcal{D}_t$ are used in training. When sufficiently learned the task in the non-dominant language $t$, the learned model should achieve good performance on the language $t$'s data $\mathcal{D}_t$ and the inaccessible data $\mathcal{D}_{unseen}$.

## 3.2 PROBLEM FORMULATION

We study the cross-lingual transfer problem from a gradient perspective. Since the goal of cross-lingual transfer is to use a dominant language to improve the accuracy of a non-dominant language in the downstream task (Lin et al., 2019), the training that decreases the loss on the dominant or mixed languages' data should also decrease the loss (i.e., improve the accuracy) on the target data $\mathcal{D}_t$ and $\mathcal{D}_{unseen}$. Training by gradient descent, this can be achieved by maximizing the inner product between the model's gradients on the losses (Riemer et al., 2019), which is our **gradient alignment** objective. By maximizing the gradient inner product, the gradient on the training data tends to be in the same direction as that on the target data and thus decreases the losses simultaneously.

Denote the training data as $\mathcal{D}_{train}$ and the model's loss on it as $\mathcal{L}_{\mathbf{w}}(\mathcal{D}_{train})$, where $\mathbf{w}$ represents the model parameters. The gradient alignment objective is to maximize the inner product between gradients on the training and non-dominant language's data:

$$\max_{\mathbf{w}} \underbrace{\nabla \mathcal{L}_{\mathbf{w}}(\mathcal{D}_{train}) \cdot \nabla \mathcal{L}_{\mathbf{w}}(\mathcal{D}_t)}_{\text{Gradient alignment to } \mathcal{D}_t} + \underbrace{\nabla \mathcal{L}_{\mathbf{w}}(\mathcal{D}_{train}) \cdot \nabla \mathcal{L}_{\mathbf{w}}(\mathcal{D}_{unseen})}_{\text{Gradient alignment to } \mathcal{D}_{unseen}}, \tag{1}$$

which includes the alignment with gradients on the accessible language $t$'s data $\mathcal{D}_t$ and the inaccessible data $\mathcal{D}_{unseen}$. In the scenario of dominant language direct transfer, $\mathcal{D}_{train} = \mathcal{D}_s$; in the scenario of mixed language transfer, $\mathcal{D}_{train} = [\mathcal{D}_s, \mathcal{D}_t]$.

**Connection to Representation Alignment** Gradient alignment depends on representation alignment, which is shown to be important for cross-lingual transfer (Li et al., 2024; Zhang et al., 2025a). We show the connection by theoretical analysis of a toy example.

In the example, we have a linear regression model with parameter $\mathbf{w} \in \mathbb{R}^d$. For any single sample input $\mathbf{x} \in \mathbb{R}^{1 \times d}$, the model outputs $\hat{y} = \mathbf{x}\mathbf{w}$. Assume there is an optimal parameter $\mathbf{w}^*$ that fits samples from the same task in both language $s$ and $t$, the loss on each sample is $||\hat{y} - \mathbf{x}\mathbf{w}^*||_2^2$.

Then the inner product of gradients on $\mathcal{D}_{train}$ and $\mathcal{D}_t$ is:

$$\nabla \mathcal{L}_{\mathbf{w}}(\mathcal{D}_{train}) \cdot \nabla \mathcal{L}_{\mathbf{w}}(\mathcal{D}_t) = \gamma \sum_{i,j} \underbrace{\mathbf{x}_t^i (\mathbf{x}_{train}^j)^T}_{\text{RepAlign 1}} \sum_m \underbrace{\lambda_m (\mathbf{x}_t^i \mathbf{q}_m)(\mathbf{x}_{train}^i \mathbf{q}_m)^T}_{\text{RepAlign 2}}, \qquad (2)$$

where $\gamma$ is a positive scalar, $\mathbf{x}_t^i \in \mathbb{R}^{1 \times d}$ is the $i$-th sample feature in $\mathcal{D}_t$, and $\mathbf{x}_{train}^j$ is the $j$-th sample feature in $\mathcal{D}_{train}$. $\mathbf{q}_m$ is an orthonormal basis of the solution space $(\mathbf{w} - \mathbf{w}^*)(\mathbf{w} - \mathbf{w}^*)^T$, with the corresponding eigenvalue $\lambda_m \geq 0$. The derivation is in Appendix A.

In Eq. (2), the inner product of gradients depends on the correlations *RepAlign1* and *RepAlign 2* between data representations (features). *RepAlign1* shows the correlation between input data representations, while *RepAlign2* shows the correlation between data representations projected to the solution space. An example of gradient alignments influenced by representation alignments is shown in Fig. 1. In Fig. 1(a), correlations between input representations $\mathbf{x}_{train}$, $\mathbf{x}_t$ and their projections to the solution

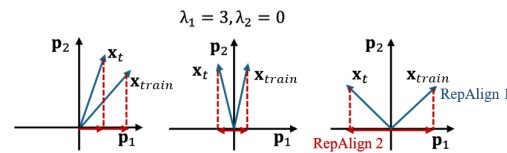

Figure 1: An example of different alignments in a 2d space. $\mathbf{p}_1$, $\mathbf{p}_2$ are orthonormal basis of the solution space with eigenvalues $\lambda_1$ and $\lambda_2$.

space are in the same direction, which causes positive gradient alignment. In Fig. 1(b), although input representations are positively correlated, their projections to the solution space $\mathbf{p}_1$ are negatively correlated, which causes negative alignment. In Fig. 1(c), $\mathbf{x}_{train} \perp \mathbf{x}_t$ and there is no gradient alignment effect.

We discuss ways to improve gradient alignments in different scenarios in the following sections.

## 4 DOMINANT LANGUAGE DIRECT TRANSFER

In the dominant language direct transfer, we train the model only on data from the dominant language (i.e. $\mathcal{D}_{train} = \mathcal{D}_s$) and then directly apply the model on the non-dominant language's data. Therefore, it is infeasible to explicitly maximize the gradient alignment between $\nabla \mathcal{L}_{\mathbf{w}}(\mathcal{D}_{train})$ and $\nabla \mathcal{L}_{\mathbf{w}}(\mathcal{D}_t)$ in Eq. (1) during training. However, considering the connection between gradient and representation alignment, LLMs with aligned pre-trained representations in language $s$ and $t$ and can still activate the gradient alignment effect. For example, if pre-trained representations of samples in $\mathcal{D}_s$ and $\mathcal{D}_t$ have large positive correlations (*RepAlign1*) when they rely on the same knowledge to solve the task (*RepAlign2*), they can still achieve positive gradient alignment even only tuning on $\mathcal{D}_s$ for the task.

We hypothesize that an LLM has aligned pre-trained representations if it has highly correlated pre-trained representations of translated data pairs, which rely on similar knowledge to solve the task. We quantify LLMs' pre-trained representation alignment ability by the deviation of cosine similarities between pre-trained token representations in and across languages (Appendix B). The smaller cosine deviations indicate better representation alignment ability. Then we evaluate the correlation between different LLMs' representation alignment ability and their direct transfer performance in Fig. 2.

In Fig. 2(c), the models' direct transfer performance tend to decrease when the cosine deviation gets large. This suggests that better representation alignment may lead to better direct transfer performance. The visualization of cosine similarities between tokens are shown in Fig. 2(a). Qwen 2.5 7B has evenly distributed cosine similarity within and across English (En) and Chinese (Zh). On the other hand, Llama 3.1 8B's cosine similarity deviation between En and Zh is larger and its direct transfer performance on Zh is also worse than Qwen. We also visualize representations of the En and Zh sentences by average pooling their token representations. Better direct transfer models (Qwen 2.5) may have close sentence representations in middle layers as well. However, different pooling strategies may influence the representation distance (Zhang et al., 2025a).

For models with good representation alignment ability, direct transfer from the dominant language may enable decent performance on non-dominant languages. This suggests improving models pre-trained representation alignment across different languages and tasks for cross-lingual transfer (Muennighoff et al., 2023; Chua et al., 2025). However, we can further improve models' alignment ability by post-training with task data in non-dominant language, discussed in the following section.

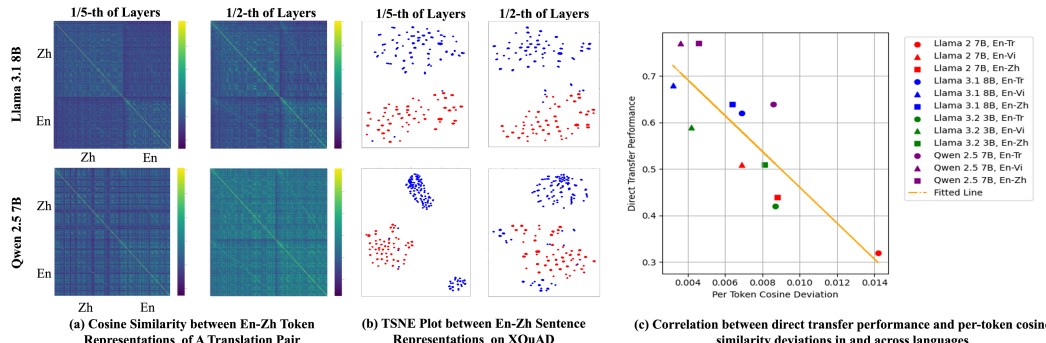

(a) Cosine Similarity between En-Zh Token Representations of A Translation Pair

(b) TSNE Plot between En-Zh Sentence Representations on XQuAD

(c) Correlation between direct transfer performance and per-token cosine similarity deviations in and across languages

Figure 2: (a) The cosine similarity between *token* representations in a En-Zh translated sentence on XQuAD; (b) The scatter of En-Zh *sentence* representations, which are computed by average pooling the token representations in sentences; (c) Correlation between LLMs' direct transfer performance (XQuAD F1) and their cosine similarity deviations between En-En and En-Zh token representations.

## 5 MIXED LANGUAGE TRANSFER

When mixed language data $\mathcal{D}_s$ and $\mathcal{D}_t$ is available in training, we can explicitly train the model to maximize the gradient alignment between language $s$ and $t$; and further enhance the representation alignment. In this section, we show that existing training strategies including target language fine-tuning and multi-task learning already improve the gradient alignment between $\mathcal{D}_{train}$ and the non-dominant data $\mathcal{D}_t$. However, they may ignore potential gradient alignment to inaccessible data $\mathcal{D}_{unseen}$ especially in the low-resource scenario. To address this issue, we use in-context prompt tuning (CL-ICP) model which utilizes the target sample's attention on mixed languages' samples to infer its relation to the unseen data $\mathcal{D}_{unseen}$. In addition, we shift the representation of the target sample to better align with the representation of the demonstration samples.

### 5.1 HOW DO EXISTING METHODS IMPROVE GRADIENT AND REPRESENTATION ALIGNMENT

**Gradient Alignment** Many works show that simply using fine-tuning (FT) and multi-task learning (MTL) already perform well in many cross-lingual transfer scenarios (Chirkova & Nikoulina, 2024; Chua et al., 2025; Gaschi et al., 2023; Wu et al., 2023; M'hamdi et al., 2021; Mousi et al., 2024). By calculating the second-order Taylor expansion of gradients (Nichol et al., 2018) in FT and MTL, we show that this is because FT and MTL improve the gradient alignment:

*Target Language Fine-tuning (FT):* In FT, we have $\mathcal{D}_{train} = \mathcal{D}_t$ and the gradient $g_{\mathbf{w}}(\mathcal{D}_{train})$ is:

$$g_{\mathbf{w}}(\mathcal{D}_{train}) = \nabla\mathcal{L}_{\mathbf{w}}(\mathcal{D}_{train}) - \frac{\beta}{2}\nabla\underbrace{\left(\nabla\mathcal{L}_{\mathbf{w}}(\mathcal{D}_t)\cdot\nabla\mathcal{L}_{\mathbf{w}}(\mathcal{D}_t)\right)}_{\text{Gradient alignment to }\mathcal{D}_t} \quad (3)$$

where $\beta$ is the learning rate times the number of gradient descent steps.

*Multi-Task Learning (MTL):* In MTL, we have $\mathcal{D}_{train} = [\mathcal{D}_s, \mathcal{D}_t]$ and the gradient $g_{\mathbf{w}}(\mathcal{D}_{train})$ is:

$$g_{\mathbf{w}}(\mathcal{D}_{train}) = \nabla\mathcal{L}_{\mathbf{w}}(\mathcal{D}_{train}) - \frac{\beta}{2}\nabla\left(\nabla\mathcal{L}_{\mathbf{w}}(\mathcal{D}_s)\cdot\nabla\mathcal{L}_{\mathbf{w}}(\mathcal{D}_s)\right)$$
$$- \frac{\beta}{2}\nabla\underbrace{\left((2\nabla\mathcal{L}_{\mathbf{w}}(\mathcal{D}_s) + \nabla\mathcal{L}_{\mathbf{w}}(\mathcal{D}_t))\cdot\nabla\mathcal{L}_{\mathbf{w}}(\mathcal{D}_t)\right)}_{\text{Gradient alignment to }\mathcal{D}_t}.$$

The derivation is in Appendix A. As shown above, gradient descent of FT and MTL tends to maximize the inner products between gradient on the training data and the data $\mathcal{D}_t$. This is a part of our gradient alignment objective in Eq. (1).

As discussed in Section 4, when sample representations correlate well after pre-training, simply using FT or MTL may achieve good gradient alignment and have decent cross-lingual performance.

Otherwise, the gradient alignment may be limited (e.g., due to small absolute value of *RepAlign 1*). This may require explicit alignment between representations.

**Representation Alignment** Previous works explicitly strengthen representation alignment to achieve better cross-lingual transfer performance (Li et al., 2024; Zhang et al., 2025b; Tang et al., 2024; Zhang et al., 2025b), which directly improves *RepAlign 1* and may make models more capable for gradient alignment. However, accurate representation alignment (e.g. via contrastive learning) usually requires rich data of translation pairs, and thus may be infeasible for non-dominant language $t$ which usually has low-resource data $\mathcal{D}_t$ (Zhang et al., 2025a). On the other hand, coarse representation alignment may cause negative gradient alignment (Fig. 1(b)) and not improve performance.

As analyzed above, existing gradient and representation alignment methods focus on the alignment between training data and accessible data $\mathcal{D}_t$, which may omit the alignment with the unseen data $\mathcal{D}_{unseen}$ especially in the low-resource scenarios (Fig. 5).

## 5.2 CROSS-LINGUAL IN CONTEXT PROMPT TUNING (CL-ICP)

In this section, we show that one can utilize LLMs' in-context learning (ICL) ability to improve the model's gradient alignment with unseen data $\mathcal{D}_{unseen}$.

Traditional ICL uses some data samples as demonstration to enhance a target sample's prediction. As shown in Table 1, we find that pre-trained LLMs allocate higher attention on demonstration samples that contribute more to the target sample's prediction. The target sample's various attention on demonstration samples

Table 1: Averaged layer attention on the demonstration (Attn) and F1 score for **un-trained** Cross-lingual ICL. -R stands for random pairs and -T stands for translation pairs. Relevant demonstrations (translation pairs) obtain more attention.

| XQuAD | En(R)-Zh | | Zh(R)-Zh | | En(T)-Zh | | Zh(T)-Zh | |
|---|---|---|---|---|---|---|---|---|
| | Attn | F1 | Attn | F1 | Attn | F1 | Attn | F1 |
| Llama3.1 | 0.73 | 0.53 | 0.74 | 0.59 | 0.78 | 0.68 | 0.82 | 0.98 |
| Qwen2.5 | 0.50 | 0.78 | 0.52 | 0.79 | 0.54 | 0.87 | 0.60 | 0.99 |

may infer its relation to unseen samples in $\mathcal{D}_{unseen}$. For example, assume that $\mathcal{D}_{train}$ does not include translation pairs in language $s$ and $t$. Then for a target sample in $\mathcal{D}_t$, demonstration samples in $\mathcal{D}_s$ with high in-context attention may be close to its translation in language $s$. In training, the target samples are supposed to have more alignment with more related demonstration samples.

**In-Context Prompt Tuning (ICP)** To incorporate ICL's alignment ability to $\mathcal{D}_{unseen}$ during training, we train for the target task by learning soft-prompts $\mathbf{p}$ with in-context samples. We learn soft prompts instead of full fine-tuning to better preserve the model's pre-trained knowledge in arranging high attention to related demonstrations.

In ICP, our model input is $\{\mathbf{p}, \mathbf{x}_d, y_d, \mathbf{x}_t\}$, where $\{\mathbf{x}_d, y_d\} = \{\mathbf{x}_{d1}, y_{d2}, ..., \mathbf{x}_{dk}, y_{dk}\}$ includes $k$ demonstration samples; $\{\mathbf{x}_t, y_t\}$ is the target sample from $\mathcal{D}_t$. We train soft prompts $\mathbf{p}$ to predict the target value $y_t$ and the demonstration value $y_{dk}$ to learn the knowledge in both languages. The objective is $\max_{\mathbf{p}} p(y_t|\{\mathbf{p}, \mathbf{x}_d, y_d, \mathbf{x}_t\}) + p(y_{dk}|\{\mathbf{p}, \mathbf{x}_{d1}, y_{d2}, ..., \mathbf{x}_{dk}\})$.

**Alignment Effect of ICP** We analyze the gradient alignment effect of ICP under the attention mechanism (Vaswani et al., 2017). For simplicity, we assume each sample representation in $\mathbf{x}_d$ has the size $\mathbb{R}^{1 \times d}$. Then the representation $\mathbf{x}$ for predicting $y_t$ is the weighted sum of in-context sample representations:

$$\mathbf{x} = \text{attn}_t(\mathbf{p})\mathbf{p} + \text{attn}_t(\mathbf{x}_d)\mathbf{x}_d + \text{attn}_t(\mathbf{x}_t)\mathbf{x}_t + \text{attn}_t(y_d)y_d,$$

where $\text{attn}_t(\cdot)$ is the self attention with the query of $\cdot$ for $y_t$ prediction. Based on Table 1, $\text{attn}_t(\cdot)$ is higher when $\cdot$ contributes more to $y_t$'s prediction.

Based on Eq. (3), by tuning for a target task with data $\mathbf{x}^i$ and $\mathbf{x}^j$, gradient descent tends to maximize the inner product between gradients $\nabla \mathcal{L}_{\mathbf{w}}(\mathbf{x}^i) \cdot \nabla \mathcal{L}_{\mathbf{w}}(\mathbf{x}^j)$. Based on the expansion of gradients in Eq. (2), $\nabla \mathcal{L}_{\mathbf{w}}(\mathbf{x}^i) \cdot \nabla \mathcal{L}_{\mathbf{w}}(\mathbf{x}^j)$ is related to the inner product between $\mathbf{x}^i$ and $\mathbf{x}^j$:

$$\mathbf{x}^i(\mathbf{x}^j)^T = \underbrace{\phi_1 \mathbf{x}_d^i (\mathbf{x}_d^j)^T + \phi_2 \mathbf{x}_d^i (\mathbf{x}_t^j)^T + \phi_3 \mathbf{x}_t^i (\mathbf{x}_t^j)^T}_{\text{Pre-trained correlation between sample representations}} + f(\{\mathbf{x}_d^{i,j}, \mathbf{x}_t^{i,j}, y_d^{i,j}\}\mathbf{p}^T), \qquad (4)$$

where $f(\{\mathbf{x}_d^{i,j}, \mathbf{x}_t^{i,j}, y_d^{i,j}\}\mathbf{p}^T)$ is the correlation between soft prompts $\mathbf{p}$ and in-context representations; $\phi_1 = \text{attn}_t(\mathbf{x}_d^i)\text{attn}_t(\mathbf{x}_d^j)$, $\phi_2 = 2\text{attn}_t(\mathbf{x}_d^i)\text{attn}_t(\mathbf{x}_t^j)$, $\phi_3 = \text{attn}_t(\mathbf{x}_t^i)\text{attn}_t(\mathbf{x}_t^j)$ are pre-trained attention correlations. We omit the non-correlation terms in the equation.

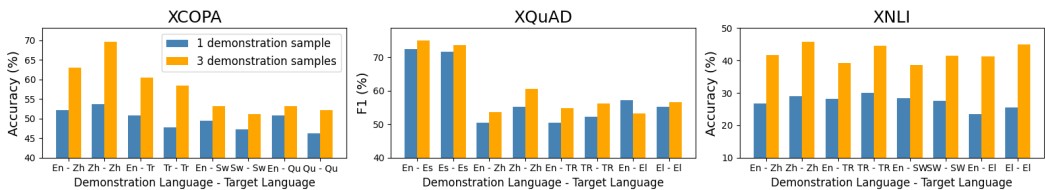

Figure 3: Llama 3.1 8B's ICL performance with different demonstration samples.

To maximize the inner product between gradients, the model may learn prompts $\mathbf{p}$ to increase the representation inner product $\mathbf{x}^i(\mathbf{x}^j)^T$. With in-context samples, such process is guided by the pre-trained correlation $\phi$ between sample representations (Eq. (4)). Without in-context samples, we have the representation $\mathbf{x} = \mathbf{x}_t$ and the inner product $\mathbf{x}^i(\mathbf{x}^j)^T = \mathbf{x}_t^i(\mathbf{x}_t^j)^T + f(\mathbf{x}_t^{i,j}\mathbf{p}^T)$, which do not include the pre-trained correlation between samples and may require sufficient data to learn.

**Mixed Language Demonstrations** According to Tanwar et al. (2023b); Zhang et al. (2024), LLMs' ICL ability highly depends on the selection of demonstration samples. In Fig. 3, we have two observations about cross-lingual ICL: (1). For a target sample in the non-dominant language $t$, using demonstration samples in language $t$ sometimes achieves better ICL performance than using English (dominant language) demonstrations. (2). Using more demonstration samples ($k = 3$) achieves better ICL performance than only using one.

Based on these observations, our CL-ICP model uses 3 demonstration samples in most tasks. Instead of choosing demonstrations from one language, we randomly select demonstration samples in $\{\mathbf{x}_d, y_d\}$ from both $\mathcal{D}_s$ and $\mathcal{D}_t$ to balance the relatedness between samples and the knowledge from the dominant language.

**Representation Shift** In Eq. (4), when pre-trained correlation between sample representations is low, CL-ICP may not well connect the demonstration and target samples and thus downgrade to single-sample prompt tuning. Therefore, we shift representations of $\mathbf{x}_t$ to further enhance its representation alignment to the demonstrations $\mathbf{x}_d$. An overview of the shifting operation is in Fig. 4.

In each layer of transformer models, the sentence inputs $\mathbf{x}_d$ and $\mathbf{x}_t$ are sequences of token representations. We denote their layer-wise token representations as $\mathbf{H}_d^L = [\mathbf{h}_{d,1}^L, ... \mathbf{h}_{d,n_d}^L]$ and $\mathbf{H}_t^L = [\mathbf{h}_{t,1}^L, ... \mathbf{h}_{t,n_t}^L]$ at the layer $L$. Inspired by Xu et al. (2023); Zhang et al. (2025b), we add representation deviations on each token representation in the target sample at the $L_{\text{to}}$ and $L_{\text{back}}$ layers where representations from different languages are close to each other. The representation deviations are calculated by

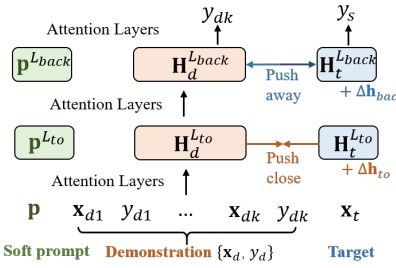

Figure 4: CL-ICP with representation shift.

$$\Delta\mathbf{h}^{L_{\text{to}}} = \text{sentpool}(\mathbf{H}_d^{L_{\text{to}}}) - \text{sentpool}(\mathbf{H}_t^{L_{\text{to}}}); \quad \Delta\mathbf{h}^{L_{\text{back}}} = \text{sentpool}(\mathbf{H}_t^{L_{\text{back}}}) - \text{sentpool}(\mathbf{H}_d^{L_{\text{back}}})$$

where sentpool calculates the average token representations as the sentence representation.

The representation shift first projects the target sample representations close to the demonstration's representation by $\Delta\mathbf{h}^{L_{\text{to}}}$ and then projects them back by $\Delta\mathbf{h}^{L_{\text{back}}}$. By making target sample representations close to demonstration representations, the model may easier find the connection between the demonstration and target samples, and thus benefit in-context prompt tuning.

# 6 EXPERIMENTS

## 6.1 SETUP

**Datasets** Our experiments are performed on three cross-lingual datasets: (1) *XQuAD*(Artetxe et al., 2019) for multilingual question answering, with Chinese (Zh), Spanish (Es), Turkish (Tr), Greek (El) as target non-dominant languages. We report exact-match and F1 score on this dataset; (2) *XCOPA* (Ponti et al., 2020) for multilingual causal reasoning, with Chinese (Zh), Swahili (Sw), Turkish (Tr), Quechua (Qu) as target non-dominant languages. We report accuracy on the dataset. (3). *XNLI* (Conneau et al., 2018) for text classification with Chinese (Zh), Swahili (Sw), Turkish (Tr), Greek

Table 2: Evaluation Results. The **bold** and underline scores are the first and second best scores.

| Dataset | XQuAD (EM/F1 %) | | | | | XCOPA (Acc %) | | | | | XNLI (Acc %) | | | | |
|---|---|---|---|---|---|---|---|---|---|---|---|---|---|---|---|
| Language $t$ | Zh | Es | Tr | El | Avg | Zh | Tr | Sw | Qu | Avg | Zh | Tr | Sw | El | Avg |
| **Llama3.1** | 58.0/63.7 | 38.2/62.5 | **33.2/62.4** | 32.8/59.7 | 40.5/62.1 | 63.2 | 54.6 | 52.6 | 52.2 | 55.7 | 61.6 | 58.3 | 51.8 | 58.7 | 57.6 |
| TL | 54.3/62.3 | 40.9/67.2 | 17.6/49.1 | 35.7/61.7 | 37.1/60.1 | 81.7 | 62.3 | 52.5 | **53.6** | 62.5 | 67.8 | 65.4 | 53.2 | 68.3 | 63.7 |
| MTL | 51.3/57.5 | 42.3/68.6 | 18.1/51.3 | 38.2/62.3 | 37.5/59.9 | 83.5 | 65.4 | 53.1 | 48.4 | 62.6 | 67.0 | 64.2 | 54.1 | 67.5 | 63.2 |
| Shifcon | 57.4/64.1 | 42.4/68.8 | 23.4/54.3 | **42.2/66.3** | 41.3/63.4 | 84.2 | 65.7 | 54.3 | 50.9 | 63.8 | 68.6 | 65.8 | 53.6 | 67.6 | 63.9 |
| CL-ICP | 53.4/65.5 | **46.4/72.3** | 25.6/58.7 | 39.1/66.1 | 41.1/65.7 | 87.6 | 77.9 | **58.8** | 49.1 | **68.4** | 70.3 | 66.1 | 59.6 | 72.0 | **67.0** |
| CL-ICP (S) | **61.6/69.1** | 44.8/71.9 | 23.1/56.3 | 41.2/**67.0** | **42.7/66.1** | **87.7** | **78.5** | 57.7 | 49.0 | 68.2 | 70.4 | 68.3 | 57.4 | 73.6 | **67.4** |
| **Qwen2.5** | 73.9/77.3 | 51.7/75.5 | 42.0/64.5 | 39.4/65.2 | 51.8/70.6 | 91.0 | 75.2 | 52.6 | 51.4 | 67.5 | 77.0 | 69.1 | 44.3 | 70.9 | 65.3 |
| TL | 78.6/**82.2** | 53.5/75.4 | 35.3/65.1 | 42.0/66.6 | 52.3/72.3 | 92.1 | 77.2 | 53.5 | 52.0 | 68.7 | 81.7 | 74.1 | 51.4 | 76.3 | 70.9 |
| MTL | 76.9/80.8 | 54.1/75.6 | 38.1/67.2 | **43.6/67.6** | 53.2/72.8 | 93.8 | 77.2 | 54.1 | 51.8 | 69.2 | 81.4 | 76.7 | 52.0 | 78.4 | 72.1 |
| Shifcon | 76.8/80.9 | **54.3/76.1** | 37.8/67.0 | 42.3/67.4 | 52.8/72.8 | 93.3 | 76.7 | 53.2 | 50.9 | 68.5 | 81.0 | 77.1 | 51.4 | 78.9 | 72.1 |
| CL-ICP | **78.7/81.4** | 51.5/75.7 | 43.7/68.2 | 40.2/66.2 | **53.5/72.9** | 93.9 | 79.6 | 55.6 | 54.0 | 70.8 | 81.5 | 77.2 | 52.6 | 77.8 | **72.3** |
| CL-ICP (S) | 77.9/80.6 | 51.5/75.2 | **44.1**/68.0 | 39.4/66.3 | 53.2/72.5 | **94.0** | **82.2** | **55.9** | 52.6 | **71.2** | 81.9 | 76.2 | **54.1** | 76.3 | 72.1 |

(El) as target non-dominant languages. We randomly sample 1000 training and testing samples for XNLI (Tu et al., 2022; Zhao et al., 2025b). For all datasets, we set English as the dominant language. The non-dominant languages are selected based on their dominance and the richness of linguistic resources (Li et al., 2024). For each dataset, we select at least one language from each dominance level (high → low).

**Models** We use Qwen 2.5 7B (Qwen et al., 2025) and Llama 3.1 8B (Dubey et al., 2024) as base models of testing methods. The methods we compare are: (1) *Dominant Language Direct Transfer (default model performance)*: only fine-tuning the pre-trained LLM on the dominant language and evaluating its performance on non-dominant language. (2) *Target Language Fine-Tuning (FT)*: fine-tuning the pre-trained LLM on the non-dominant language. (3) *Transfer Learning (TL)*: first fine-tuning the LLM on the dominant language and then fine-tuning on the non-dominant language. (4) *Multi-Task Learning (MTL)*: Fine-tuning the model on both dominant and non-dominant languages' data. (5) *ShifCon* (Zhang et al., 2025a): MTL model with representation projection and contrastive learning to align sample representations. (6) *CL-ICP*: our cross-lingual in-context prompt tuning without representation shift. (7) *CL-ICP (S)*: our CL-ICP model with representation shift. This is a short term of CL-ICP (Shift).

For ICP models, we use demonstration samples in the training set at test time for a fair comparison with other methods. Due to the input length constraint, we use 1 demonstration samples for XQuAD and 3 demonstration samples for other datasets. For every method, we compute the scores averaged from three random seeds. Detailed training settings are in the appendix.

## 6.2 EXPERIMENTAL RESULTS

**Results of the Gradient Alignment** We show results of different methods in Table 2. CL-ICP (with or without shift) achieves the best average performance across all datasets. The effectiveness of CL-ICP varies between tasks, base models and languages.

For the question answering task XQuAD which may have sequence-to-sequence distribution similar to pre-trained data, the direct transfer already achieves decent performance and the improvement of CL-ICP is marginal. For other tasks XCOPA and XNLI whose outputs are choices instead of sentences, the direct transfer is not effective enough and CL-ICP achieves much better performance compared to other baselines.

Comparing performance on different base models, CL-ICP outperforms baselines by larger margins on Llama 3.1 than Qwen 2.5. This may be because Qwen 2.5 has better pre-trained representation alignment than Llama 3.1 (as shown in Fig. 2(c)), which makes simple fine-tuning on two languages (e.g. MTL) strong baselines.

Relying on pre-trained LLMs' ICL ability, CL-ICP's performance also depends on LLMs pre-trained capacity on different languages. For the rare language that are poorly learned in pre-training (e.g., Qu in Llama 3.1, El in Qwen 2.5), CL-ICP does not outperform baselines with full fine-tuning. For El in Qwen 2.5, the model may need full fine-tuning instead of prompt tuning to learn the knowledge in the non-dominant language. For Qu in Llama 3.1, the model may have difficulty learning the relationship between En and Qu, which makes the two-stage training (direct transfer and TL) performs best.

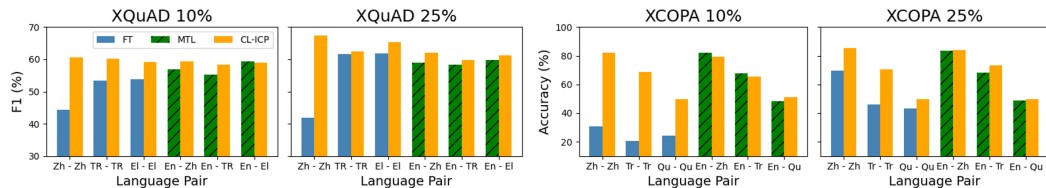

Figure 5: Llama 3.1 8B's performance with different ratios of non-dominant language data. Under the language pair $s - t$, we compare CL-ICP with mixed language demonstrations and MTL; under the language pair $t - t$, we compare CL-ICP with non-dominant language demonstrations and FT.

**Results of the Representation Alignment** In Table 2, we compare the models with additional representation alignment, ShifCon and CL-ICP (Shift). With contrastive learning on shifted representations, ShifCon outperforms MTL in most cases on Llama 3.1. With shifted representations, CL-ICP (Shift) also outperforms CL-ICP in 2/3 datasets on Llama 3.1.

However, the effect of representation alignment is critical in different scenarios. On Qwen 2.5, Shifcon does not outperform MTL in some datasets and languages; CL-ICP (Shift)'s performance also differs in different base models and datasets. That may be because representation alignment needs carefully designed strategies and sufficient data for training (Zhang et al., 2025a). By only learning on the target tasks' data, ShifCon and CL-ICP (Shift) may not learn the true relationship between shifted representations of data in different languages. As shown in Fig 1(b), this may cause negative gradient alignment and even decrease the performance.

**Results in Low Resource Scenarios** We show results of training with different ratios of non-dominant language data in Fig. 5. We use full dominant language (English) data in this experiment. Results show that with low-resource non-dominant language data, FT suffers from the generalization issue which makes it underform MTL by a large margin (with less non-dominant language data, MTL is closer to the direct transfer). However, using the same data, CL-ICP significantly outperform FT, which shows its generalization ability in aligning with unseen data.

However, when the demonstration samples are in mixed languages, CL-ICP needs sufficient data to address the relationship between demonstration and target samples. With 10% data in XQuAD and XCOPA, CL-ICP with mixed language demonstrations performs worse than MTL in some cases and CL-ICP with non-dominant language demonstrations achieves better performance. With more than 25% non-dominant data, CL-ICP consistently outperform MTL.

## 6.3 ABLATION STUDY

**Influence of Demonstration** The experiments of ablation studies use Llama 3.1 8B as the base model. We show the influence of the number and mixture of demonstration samples in Table 3. Using 3 demonstration samples in mixed languages achieves best performance. Using 3 demonstration samples in the target non-dominant language outperforms that in the dominant language (English). In addition, using mixed language demonstration with only 1 sample sometimes outperforms using English demonstration with 3 samples. These indicate the importance of close distribution between demonstration and target samples in in-context training.

Table 3: CL-ICP with different demonstrations on XCOPA.

| Lang, # | Zh | Tr | Sw | Qu | Avg |
|---|---|---|---|---|---|
| English, 3 | 86.0 | 74.4 | 55.9 | 50.1 | 66.6 |
| Target, 3 | 87.3 | 77.1 | 56.7 | **50.9** | 68.0 |
| Mixed, 1 | 83.7 | 75.7 | 57.9 | 49.7 | 66.8 |
| Mixed, 3 | **87.6** | **77.9** | **58.8** | 49.1 | **68.4** |

**Influence of In-Context Inference** For a fair comparison with our model, one question is: will baseline models achieve better performance when adding the same in-context samples as CL-ICP at the inference time? We show the results in Table 4. Results show that adding in-context samples will improve baselines' performance in some cases, but still underperform our CL-ICP model.

Table 4: Models with in-context samples at inference time on XCOPA.

| Model | Zh | Tr | Sw | Qu | Avg |
|---|---|---|---|---|---|
| TL | 81.7 | 62.3 | 52.5 | **53.6** | 62.5 |
| + ICL | 78.3 | 67.2 | 53.3 | 50.5 | 62.3 |
| MTL | 83.5 | 65.4 | 53.1 | 48.4 | 62.6 |
| + ICL | 80.5 | 70.0 | 55.0 | 52.0 | 64.4 |
| CL-ICP | **87.6** | **77.9** | **58.8** | 49.1 | **68.4** |

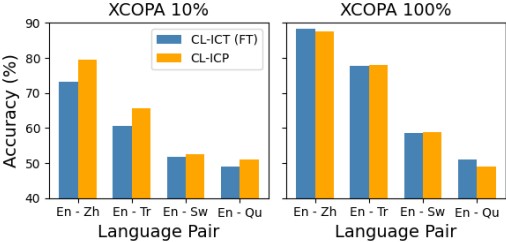

Figure 6: Comparison between CL-ICP and in-context fine-tuning (CL-ICT).

Table 5: Multi-task learning (MTL) and cross-lingual in-context tuning (CL-ICT) by only training prompts (PT) or LoRA blocks.

|  |  | Zh | Tr | Sw | Qu | Avg |
|---|---|---|---|---|---|---|
| PT | MTL | 56.4 | 55.8 | 52.6 | 50.6 | 53.8 |
|  | CL-ICT | 87.6 | 77.9 | 58.8 | 49.1 | 68.4 |
| LoRA | MTL | 87.4 | 75.6 | 59.7 | 50.0 | 68.2 |
|  | CL-ICT | 88.2 | 81.5 | 63.3 | 53.1 | **71.5** |

**Influence of Prompt Tuning** We compare the influence of prompt tuning and full parameter tuning in our cross-lingual in-context training in Fig. 6. Although in-context fine-tuning achieves slightly better performance with 100% non-dominant language's data, it underperforms CL-ICP in the low-resource scenario. This may be because in the low-resource scenario, full parameter tuning may be easier to lose alignment with unseen data due to the distortion pre-trained sample representations.

We also compare the in-context training using LoRA and prompt tuning with MTL in Table 5. In practice, prompt tuning only may be hard to train and the model may have low capacity in solving the target tasks. Using LoRA can improve the model's performance compared to full finetuning. However, such a benefit is also applicable to our in-context training model. Using LoRA, our model still achieves better performance than MTL. This suggests that our improvements come not only from the parameter-efficient training, but also from the inclusion of in-context samples during training.

## 7 CONCLUSION

In this paper, we presented a comprehensive study on the cross-lingual transfer in large language models, with a particular focus on the interplay between representation and gradient alignment. To improve cross-lingual transfer in post-training, we propose a cross-lingual in-context prompt tuning model to improve gradient alignment and add a representation shift to better align representations between demonstration and target samples. Experiments show that our models have improved performance in both low and high resource scenarios. The limitation of our model is that it depends on LLMs' ICL capacity. When adding the demonstration in the input, the input sequence length will increase and is inefficient for long samples. We leave these questions for future study.

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

# A  DERIVATION

## A.1  CONNECTION BETWEEN GRADIENT ALIGNMENT AND REPRESENTATION ALIGNMENT

This is the detailed derivation of Eq. (2).

In the example, we have a linear regression model with parameter $\mathbf{w} \in \mathbb{R}^d$. For the task in language $s$ and $t$, we have data $\mathcal{D}_{train} = \{\mathbf{X}_{train} \in \mathbb{R}^{n_{train} \times d}, \mathbf{y}_{train} \in \mathbb{R}^{n_s}\}$ and $\mathcal{D}_t = \{\mathbf{X}_t \in \mathbb{R}^{n_t \times d}, \mathbf{y}_t \in \mathbb{R}^{n_s}\}$ respectively, containing data features and the true outputs. In different transfer scenarios, $\mathcal{D}_{train}$ contains different mixes of $\mathcal{D}_s$ and $\mathcal{D}_t$. Since samples in different languages are for the same task, we assume that there is an optimal parameter $\mathbf{w}^*$ that fits both $\mathcal{D}_{train}$ and $\mathcal{D}_t$. Then for any single sample $\{\mathbf{x} \in \mathbb{R}^{1 \times d}, y \in \mathbb{R}\}$, the model outputs $\hat{y} = \mathbf{x}\mathbf{w}$ and the loss on the sample is $||\hat{y} - \mathbf{x}\mathbf{w}^*||_2^2$.

The inner product of gradients on $\mathcal{D}_s$ and $\mathcal{D}_t$ is:

$$\nabla\mathcal{L}_{\mathbf{w}}(\mathcal{D}_t) \cdot \nabla\mathcal{L}_{\mathbf{w}}(\mathcal{D}_{train})$$
$$=\gamma(\mathbf{w} - \mathbf{w}^*)^T \mathbf{X}_t^T \mathbf{X}_t \mathbf{X}_{train}^T \mathbf{X}_{train}(\mathbf{w} - \mathbf{w}_j^*)$$
$$=\gamma\sum_{i,j} \mathbf{x}_t^i(\mathbf{x}_{train}^j)^T \left[\mathbf{x}_t^i(\mathbf{w} - \mathbf{w}^*)(\mathbf{w} - \mathbf{w}^*)^T(\mathbf{x}_{train}^j)^T\right] \quad (5)$$

where $\gamma = \frac{1}{n_t n_{train}}$, $\mathbf{x}_t^i \in \mathbb{R}^{1 \times d}$ is the $i$-th row of the matrix $\mathbf{X}_t$ (i.e., the input feature of the $i$-th sample in language $t$), and $\mathbf{x}_s^j$ is the $j$-th row of the matrix $\mathbf{X}_s$.

We further decompose $(\mathbf{w} - \mathbf{w}^*)(\mathbf{w} - \mathbf{w}^*)^T = \mathbf{Q}\Sigma\mathbf{Q}^T$ where each column $m$ of $\mathbf{Q}$ is an orthonormal vector $\mathbf{q}_m$ with the corresponding eigenvalue $\lambda_m \geq 0$ as the $m$-th diagonal element in $\Sigma$. Then Eq. (5) can be written as

$$\text{Eq. (5)} =\gamma\sum_{i,j} \mathbf{x}_t^i(\mathbf{x}_{train}^j)^T \left[\mathbf{x}_t^i\mathbf{Q}\Sigma\mathbf{Q}^T(\mathbf{x}_{train}^j)^T\right]$$
$$=\gamma\sum_{i,j} \underbrace{\mathbf{x}_t^i(\mathbf{x}_{train}^j)^T}_{\text{RepAlign 1}} \sum_m \underbrace{\lambda_m\mathbf{x}_t^i\mathbf{q}_m\mathbf{q}_m^T(\mathbf{x}_{train}^j)^T}_{\text{RepAlign 2}},$$

which is Eq. (2) in the main paper.

## A.2  FT AND MTL FOR GRADIENT ALIGNMENT

This is the detailed derivation of gradient alignment effect in the FT and MTL.

### A.2.1  TARGET LANGUAGE FINE-TUNING

For a model with parameter $\mathbf{w}$, and data $\mathcal{D}_{train} = \mathcal{D}_t$ from languages $t$, the objective of Target Language Fine-Tuning is to minimize the loss $L_{\mathbf{w}}(\mathcal{D}_t)$. When updating the model by gradient descent, the parameter $\mathbf{w}$ is updated as

$$\mathbf{w}_k = \mathbf{w}_0 - \beta\sum_k L'_{\mathbf{w}_{k-1}}(\mathcal{D}_t),$$

where $\beta$ is the learning rate, $\mathbf{w}_0$ is the initialized weight of $\mathbf{w}$, $k$ is the updating step.

The gradient of target language fine-tuning at the step $k$ is:

$$g_{\mathbf{w}_k}(\mathcal{D}_t) = L'_{\mathbf{w}_k}(\mathcal{D}_t) \quad (6)$$

We now take the second-order Taylor expansion of the gradient:

$$L'_{\mathbf{w}_k}(\mathcal{D}_t) = L'_{\mathbf{w}_0}(\mathcal{D}_t) + L''_{\mathbf{w}_0}(\mathcal{D}_t)(\mathbf{w}_k - \mathbf{w}_0) + \mathcal{O}(||\mathbf{w}_k - \mathbf{w}_0||^2)$$
$$= L'_{\mathbf{w}_0}(\mathcal{D}_t) + L''_{\mathbf{w}_0}(\mathcal{D}_t)(\mathbf{w}_k - \mathbf{w}_0) + \mathcal{O}(\beta^2)$$
$$= L'_{\mathbf{w}_0}(\mathcal{D}_t) - \beta L''_{\mathbf{w}_0}(\mathcal{D}_t)\sum_k L'_{\mathbf{w}_{k-1}}(\mathcal{D}_t).$$

We omit the term $\mathcal{O}(\beta^2)$ here since the learning rate is usually small for LLMs (e.g., 1e-5 for fine-tuning). Using the Taylor expansion of $L'_{\mathbf{w}_{k-1}}(\mathcal{D}_t)$, we have

$$\beta L'_{\mathbf{w}_{k-1}}(\mathcal{D}_t) = \beta L'_{\mathbf{w}_0}(\mathcal{D}_t) + \underbrace{\beta \mathcal{O}(||\mathbf{w}_k - \mathbf{w}_0||)}_{\mathcal{O}(\beta^2)}$$

The latter $\mathcal{O}(\beta^2)$ is omitted. Then the gradient in Eq. (6) is:

$$g_{\mathbf{w}_k}(\mathcal{D}_t) = L'_{\mathbf{w}_0}(\mathcal{D}_t) - \beta k L''_{\mathbf{w}_0}(\mathcal{D}_t) L'_{\mathbf{w}_0}(\mathcal{D}_t)$$

Based on the product rule of derivatives, we have

$$g_{\mathbf{w}_k}(\mathcal{D}_t) = L'_{\mathbf{w}_0}(\mathcal{D}_t) - \frac{\beta k}{2} \nabla_{\mathbf{w}_0} \left( L'_{\mathbf{w}_0}(\mathcal{D}_t) L'_{\mathbf{w}_0}(\mathcal{D}_t) \right)$$

$$= L'_{\mathbf{w}_0}(\mathcal{D}_{train}) - \frac{\beta k}{2} \nabla_{\mathbf{w}_0} \left( L'_{\mathbf{w}_0}(\mathcal{D}_t) L'_{\mathbf{w}_0}(\mathcal{D}_t) \right) \qquad (\mathcal{D}_{train} = \mathcal{D}_t)$$

This suggests that the gradient of multi-task learning encourages to maximize the inner product of gradient on data within samples of $\mathcal{D}_t$.

### A.2.2 MULTI-TASK LEARNING

For a model with parameter $\mathbf{w}$, and data $\mathcal{D}_{train} = [\mathcal{D}_s, \mathcal{D}_t]$ from languages $s$ and $t$, we have the training loss $L_{\mathbf{w}}(\mathcal{D}_s)$ and $L_{\mathbf{w}}(\mathcal{D}_t)$, respectively. The objective of multi-task learning is to minimize the loss:

$$L_{\mathbf{w}}(\mathcal{D}_s + \mathcal{D}_t) = L_{\mathbf{w}}(\mathcal{D}_s) + L_{\mathbf{w}}(\mathcal{D}_t).$$

Following the toy example, we assume the parameter $\mathbf{w}$ is a $d$-dimensional vector. When updating the model by gradient descent, the parameter $\mathbf{w}$ is updated as:

$$\mathbf{w}_k = \mathbf{w}_0 - \beta \sum_k \left( L'_{\mathbf{w}_{k-1}}(\mathcal{D}_s) + L'_{\mathbf{w}_{k-1}}(\mathcal{D}_t) \right)$$

where $\mathbf{w}_0$ is the initialized weight of $\mathbf{w}$, $k$ is the updating step, $\beta$ is the learning rate. The gradient of multi-task learning at the step $k$ is:

$$g_{\mathbf{w}_k}(\mathcal{D}_s + \mathcal{D}_t) = L'_{\mathbf{w}_k}(\mathcal{D}_s) + L'_{\mathbf{w}_k}(\mathcal{D}_t) \qquad (7)$$

We now take the second-order Taylor expansion of each gradient. For the gradient on $\mathcal{D}_s$ we have

$$L'_{\mathbf{w}_k}(\mathcal{D}_s) = L'_{\mathbf{w}_0}(\mathcal{D}_s) + L''_{\mathbf{w}_0}(\mathcal{D}_s)(\mathbf{w}_k - \mathbf{w}_0) + \mathcal{O}(||\mathbf{w}_k - \mathbf{w}_0||^2)$$

$$= L'_{\mathbf{w}_0}(\mathcal{D}_s) + L''_{\mathbf{w}_0}(\mathcal{D}_s)(\mathbf{w}_k - \mathbf{w}_0) + \mathcal{O}(\beta^2)$$

$$= L'_{\mathbf{w}_0}(\mathcal{D}_s) - \beta L''_{\mathbf{w}_0}(\mathcal{D}_s) \sum_k \left( L'_{\mathbf{w}_{k-1}}(\mathcal{D}_s) + L'_{\mathbf{w}_{k-1}}(\mathcal{D}_t) \right).$$

We omit the term $\mathcal{O}(\beta^2)$ here since the learning rate is usually small for LLMs (e.g., 1e-5 for fine-tuning). Then since $L'_{\mathbf{w}_{k-1}}(\mathcal{D}_s) = L'_{\mathbf{w}_0}(\mathcal{D}_s) + \mathcal{O}(\beta)$, we further approximate $L'_{\mathbf{w}_k}(\mathcal{D}_s)$ as:

$$L'_{\mathbf{w}_k}(\mathcal{D}_s) = L'_{\mathbf{w}_0}(\mathcal{D}_s) - \beta k L''_{\mathbf{w}_0}(\mathcal{D}_s) \left( L'_{\mathbf{w}_0}(\mathcal{D}_s) + L'_{\mathbf{w}_0}(\mathcal{D}_t) \right).$$

Similarly, we have

$$L'_{\mathbf{w}_k}(\mathcal{D}_t) = L'_{\mathbf{w}_0}(\mathcal{D}_t) - \beta k L''_{\mathbf{w}_0}(\mathcal{D}_t) \left( L'_{\mathbf{w}_0}(\mathcal{D}_s) + L'_{\mathbf{w}_0}(\mathcal{D}_t) \right).$$

Then we expand Eq. (7) as:

$$\begin{aligned}
g_{\mathbf{w}_k}(\mathcal{D}_s + \mathcal{D}_t) = {} & L'_{\mathbf{w}_0}(\mathcal{D}_s) + L'_{\mathbf{w}_0}(\mathcal{D}_t) \\
& - \beta k L''_{\mathbf{w}_0}(\mathcal{D}_s) L'_{\mathbf{w}_0}(\mathcal{D}_s) \\
& - \beta k L''_{\mathbf{w}_0}(\mathcal{D}_t) L'_{\mathbf{w}_0}(\mathcal{D}_t) \\
& - \beta k \left( L''_{\mathbf{w}_0}(\mathcal{D}_s) L'_{\mathbf{w}_0}(\mathcal{D}_t) + L''_{\mathbf{w}_0}(\mathcal{D}_t) L'_{\mathbf{w}_0}(\mathcal{D}_s) \right)
\end{aligned}$$

Based on the product rule of derivatives, we integrate the terms in the above gradients as below.

$$L''_{\mathbf{w}_0}(\mathcal{D}_s)L'_{\mathbf{w}_0}(\mathcal{D}_s) = \frac{1}{2}L''_{\mathbf{w}_0}(\mathcal{D}_s)L'_{\mathbf{w}_0}(\mathcal{D}_s) + \frac{1}{2}L''_{\mathbf{w}_0}(\mathcal{D}_s)L'_{\mathbf{w}_0}(\mathcal{D}_s)$$

$$= \frac{1}{2}\nabla_{\mathbf{w}_0}\left(L'_{\mathbf{w}_0}(\mathcal{D}_s)L'_{\mathbf{w}_0}(\mathcal{D}_s)\right)$$

$$L''_{\mathbf{w}_0}(\mathcal{D}_t)L'_{\mathbf{w}_0}(\mathcal{D}_t) = \frac{1}{2}\nabla_{\mathbf{w}_0}\left(L'_{\mathbf{w}_0}(\mathcal{D}_t)L'_{\mathbf{w}_0}(\mathcal{D}_t)\right)$$

$$L''_{\mathbf{w}_0}(\mathcal{D}_s)L'_{\mathbf{w}_0}(\mathcal{D}_t) + L''_{\mathbf{w}_0}(\mathcal{D}_t)L'_{\mathbf{w}_0}(\mathcal{D}_s) = \nabla_{\mathbf{w}_0}\left(L'_{\mathbf{w}_0}(\mathcal{D}_s)L'_{\mathbf{w}_0}(\mathcal{D}_t)\right)$$

Then the gradient of multi-task learning is finally expanded as:

$$g_{\mathbf{w}_k}(\mathcal{D}_s + \mathcal{D}_t) = L'_{\mathbf{w}_0}(\mathcal{D}_{train}) \qquad\qquad (L'_{\mathbf{w}_0}(\mathcal{D}_{train}) = L'_{\mathbf{w}_0}(\mathcal{D}_s) + L'_{\mathbf{w}_0}(\mathcal{D}_t))$$

$$- \frac{\beta k}{2}\nabla_{\mathbf{w}_0}\left(L'_{\mathbf{w}_0}(\mathcal{D}_s)L'_{\mathbf{w}_0}(\mathcal{D}_s)\right)$$

$$- \frac{\beta k}{2}\nabla_{\mathbf{w}_0}\left(L'_{\mathbf{w}_0}(\mathcal{D}_t)L'_{\mathbf{w}_0}(\mathcal{D}_t)\right)$$

$$- \beta k\nabla_{\mathbf{w}_0}\left(L'_{\mathbf{w}_0}(\mathcal{D}_s)L'_{\mathbf{w}_0}(\mathcal{D}_t)\right)$$

This suggests that the gradient of multi-task learning encourages to maximize the inner product of gradient on data within and across languages.

# B  EXPERIMENTAL SETTINGS

## B.1  METRIC FOR COSINE DEVIATION

We hypothesize that an LLM has aligned representations if it has highly correlated representations of translated data pairs, which rely on similar knowledge to solve the task. Therefore, we compare the cosine similarity of representations between translated data pairs and shuffled data in the dominant language. Since translated pairs rely on similar knowledge to solve the task, they are expected to have higher cosine similarity than shuffled data in the dominant language.

We use the cosine deviation below to quantify LLMs' pre-trained representation alignment ability. For the $L$-th transformer layer, we have pre-trained data representations of each sentence pair in language $s$ and $t$ as $\mathbf{H}_s^L = [\mathbf{h}_{s,1}^L, ...\mathbf{h}_{s,n_s}^L]$ and $\mathbf{H}_t^L = [\mathbf{h}_{t,1}^L, ...\mathbf{h}_{t,n_t}^L]$ where $n_s$ and $n_t$ are numbers of tokens in each sentence; $\mathbf{h}_{t,n_t}^L$ and $\mathbf{h}_{s,n_s}^L$ are token representations in the corresponding sentence. The cosine deviation of all sentence pairs is computed as:

$$\text{cos\_dev}(\mathbf{H}_s^L, \mathbf{H}_t^L) = \frac{1}{n_s}\sum_i \cos(\mathbf{h}_{s,i}^L, \mathbf{H}_r^L; \text{top-10}) - \cos(\mathbf{h}_{s,i}^L, \mathbf{H}_t^L; \text{top-10})$$

$$\text{cos\_dev}(\mathbf{H}_s, \mathbf{H}_t) = \mathbb{E}[\frac{1}{L}\sum_L \text{cos\_dev}(\mathbf{H}_s^L, \mathbf{H}_t^L)]$$

where $\mathbf{H}_r^L$ is the random sentence selected from language $s$. $\cos(\mathbf{h}_{s,i}^L, \mathbf{H}_t^L; \text{top-10})$ is the average of the top-10 highest cosine similarities between the token representation $\mathbf{h}_{s,i}^L$ and all token representations in $\mathbf{H}_t^L$. Since the translated sentences in different languages may have varied token numbers due to different tokenization, we pick top-10 highest cosine similarities and select language pairs with comparable token numbers in our experiment.

## B.2  TRAINING DETAILS IN SECTION 6

We train our models on 4 Nvidia L40S machines. The detailed training settings for each model is:

For each baseline model, we select training epochs from the set $\{1, 2, 3, 4, 5, 8\}$ and pick the epoch with stable top performance.

For parameter efficient test in Table 5, we use LoRA with learning rate 2e-4, training batch 8 with 3 epochs training on 1 Nvidia A100 machine. For MTL prompt tuning, we select best results from the

| Model | LR | Batch Size | XQuAD | XCOPA | XNLI |
|-------|-----|-----------|-------|-------|------|
| TL | 2e-6 | 4 per-device | Epoch 1 on each single language | Epoch 3 on each single language | Epoch 3 on each single language |
| MTL | 2e-6 | 4 per-device | Epoch 2 | Epoch 3 | Epoch 3 |
| ShifCon | 2e-6 | 4 per-device | Epoch 2 | Epoch 4 | Epoch 4 |
| CL-ICP | 1e-3 | 4 per-device | Epoch 5, prompt 1, demosample 1 | Epoch 20, prompt 10, demosample 3 | Epoch 20, prompt 10, demosample 3 |

following settings: the same prompt setting as our CL-ICP; increasing the prompt number (20, 30, 50) and training epochs (10, 20, 30); adjusting the learning rate from (2e-4, 1e-3, 5e-3, 1e-2). The prompt embeddings are initialized the same as CL-ICP.

## C  INFLUENCE OF DEMONSTRATION TRAINING

Table 6: CL-ICL models without training to predict demonstration samples.

| Dataset | XQuAD (EM/F1 %) | | | | | XCOPA (Acc %) | | | | | XNLI (Acc %) | | | | |
|---------|------|------|------|------|------|------|------|------|------|------|------|------|------|------|------|
| Language $t$ | Zh | Es | Tr | El | Avg | Zh | Tr | Sw | Qu | Avg | Zh | Tr | Sw | El | Avg |
| CL-ICP | 53.4/65.5 | 46.4/72.3 | 25.6/58.7 | 39.1/66.1 | 41.1/65.7 | 87.6 | 77.9 | **58.8** | 49.1 | **68.4** | 70.3 | 66.1 | **59.6** | 72.0 | 67.0 |
| - Demotrain | 64.4/68.9 | **47.3/72.6** | 28.4/58.5 | 40.8/66.5 | 45.2/66.6 | **88.3** | 77.9 | 55.2 | **50.7** | 68.0 | 69.7 | 65.8 | 55.7 | 71.0 | 65.5 |
| CL-ICP (S) | 61.6/69.1 | 44.8/71.9 | 23.1/56.3 | 41.2/67.0 | 42.7/66.1 | 87.7 | 78.5 | 57.7 | 49.0 | 68.2 | **70.4** | **68.3** | 57.4 | **73.6** | **67.4** |
| - Demotrain | **65.4/69.3** | 45.5/71.2 | **31.4/60.1** | **42.0/67.0** | **46.1/66.9** | **88.3** | **80.1** | 54.2 | 50.1 | 68.2 | 70.0 | 64.9 | 53.4 | 71.0 | 64.8 |

We show the influence of training with the demonstration prediction loss in CL-ICP in Table 6. For the question answering task XQuAD which has sequence to sequence distribution close to pre-training data, adding the demonstration loss does not increase the performance. However, for task XCOPA and XNLI whose outputs are choices instead of sequences, adding the demonstration loss achieves better performance. We hypothesize that this is because the model can easily acquire knowledge from demonstration samples in XQuAD, while it needs further training to acquire knowledge of the task in XCOPA and XNLI. So for tasks which have dissimilar distribution to pre-trained data, training with the demonstration loss is beneficial.

