# OpenReview forum: "Rethinking Alignment in Cross-Lingual Knowledge Transfer"
_ICLR.cc/2026/Conference — ICLR 2026 Conference Withdrawn Submission_

### Official Review · Reviewer_5zwr · 2025-10-30

**Soundness:** 1
**Presentation:** 2
**Contribution:** 2
**Rating:** 2
**Confidence:** 3

**Summary:**

The paper studies cross-lingual transfer in LLMs via representation alignment, that is, if the LLM encodes the tokens/sentences of two languages invariantly, then training/prompting the LLM on data from one language will also makes it perform will on other languages.

Based on the reviewer's reading of the paper, the authors:

1. Relate transfer performance to the alignment of the loss gradient
    - This connection is discussed in some domain generalization literature such as "Invariant Risk Minimization".
2. On a toy linear regression example (where each example is a vector), show that gradient alignment is equal to the sum of the dot products between examples from one language and those from another language
    - Perhaps a loose connection to LLM is to take each example as the penultimate layer sentence embeddings (e.g., mean-pooled)
3. They compute and show token-level dot products between a translation pair on two LLMs in fig. 2, and show that this dot product correlates with downstream classification performance (after fine-tuning on source/dominant language only)
4. They propose CL-ICP: on top of few-show prompting with source/dominant and target/non-dominant language examples, they perform soft prompt-tuning w.r.t. both source and target languages' labeled examples.
5. On top of CL-ICL, they also propose modifying the intermediate pooled representation of the test (in another language) sequence to so that it is closer to the source language.

**Strengths:**

The reviewer appreciates effort on providing an (toy theoretical and empirical) analysis for justifying philosophy of representation alignment.

**Weaknesses:**

1. In a nutshell, CL-ICP is multi-lingual few-shot prompting + soft prompt tuning.
- As an empirical method, it is a relatively common and well-known technique, and by itself is not very surprising nor novel.
- It is compared to full-parameter fine-tuning (TL/MTL).  The reviewer is curious whether the improvement is simply due to the fact that prompt-tuning is parameter efficient (better generalization properties)?  A comparison to LoRA and simple prompt-tuning would be illustrating.

2. The connection from prompt-tuning to representation alignment eq. (4), to the toy example in eq. (2), to loss gradient alignment in eq. (1), then finally to transfer performance (IRM), is very tenuous.  The reviewer does not find the argument to be theoretically rigorous, nor well-supported empirically.
- It is unclear whether CL-ICP *actually* improves transfer performance via the hypothesized mechanism, empirically.  (Given that the lack of theoretical justification, including the final link between IRM and performance).
- Fig. 2 evaluates the correlation between representation alignment and performance AFTER fine-tuning.  By the phenomenon of "neural collapse", this correlation between alignment and performance is expected on trained neural models in general.  Instead, the reviewer is curious whether this correlation also holds for CL-ICP, and prompting-only (no tuning).

**Questions:**

See above.

---

> ### Author Response · Authors · 2025-11-20
>
> We thank the reviewer for experimental suggestions. We address each point below.
>
> **W1.1 Novelty of the paper**
>
> The novelty of our paper includes but is not limited to the CL-ICP model structure.
>
> * We consider our main novelty is to provide a new and unified perspective to study the alignment in cross-lingual knowledge transfer, as reviewer 21p8 mentioned. **In comparison, previous works mainly focus on the representation alignment. However, why and how representation alignment is necessary for cross-lingual transfer are underexplored; and it is unclear why simple training strategies can already perform well in many cases (do they have alignment ability?)**. Our work addresses these important questions.
>
> * Based on the perspective, CL-ICP is designed to improve both gradient and representation alignment for cross-lingual knowledge transfer. We show the distinction between CL-ICP and previous training strategies (FT and MTL) in addressing the gradient alignment in Section 5; and provide a novel way to add representation shifts in the in-context training. **To the best of our knowledge, the gradient alignment effect and the representation shift in the in-context training has not been studied before. Moreover, the improved performance of CL-ICP further validates our novel perspective of alignment in the paper**.
>
> **W1.2 Comparison to parameter efficient tuning**
>
> We add experiments of using full fine-tuning (FT), prompt tuning (PT) and LoRA with MTL and CL-ICP on XCOPA data below (based on Llama 3.1 8B).
>
> | |En-Zh|En-Tr|En-Sw|En-Qu|Avg|
> |---|---|---|---|---|---|
> |MTL (FT)|83.5|65.4 |53.1|48.4|62.6|
> |CL-ICP (FT)|88.3|77.7 |58.7|50.9|68.9|
> |MTL (PT)|56.4|55.8|52.6|50.6|53.8|
> |CL-ICP (PT)|87.6|77.9 |58.8|49.1|68.4|
> |MTL (LoRA)|87.4|75.6|59.7|50.0|68.2|
> |CL-ICP (LoRA)|88.2|81.5|63.3|53.1|71.5|
> * First, using full parameter in-context tuning does not degrade our performance compared to using prompt tuning.
> * Second, in practice, prompt tuning only may be hard to train. We show results of using prompt tuning without in-context samples, and the model turned out to have low capacity in solving the tasks. The results in the table are the best from the settings in Appendix B.2 line 971-980. Our hypothesis is that although prompt tuning may have generalization effects, reducing the training parameters may also degrade the model's transfer ability.
> * Third, we compare the performance using LoRA. Using LoRA can improve the model’s performance compared to full-finetuning (which may be because of the generalization benefit). However, such a benefit is also applicable to our in-context training model. Using LoRA, our CL-ICP still achieves better performance than MTL.
>
> These suggest that our improvements come not only from the parameter-efficient training. We also add the above results in our revision Table 5.
>
> **W2.1 Theoretical framework**
>
> First, we would like to clarify that our theoretical framework is based on a gradient alignment objective, which is also used to characterize the model’s transfer ability across tasks in [1]. Although it might be connected, **we do not propose claims related to IRM in the paper**.
>
> * **How CL-ICP connects to gradient alignment**: CL-ICP connects to gradient alignment by using in-context samples during training, not by prompt tuning. The connection is based on the analysis that simple fine-tuning on non-dominant language improves gradient alignment within the non-dominant language data (Eq. (3)). When using in-context samples, the input $\mathbf x$ incorporates data representation from both dominant and non-dominant languages. Based on the gradient alignment form of our toy example in Eq. (2), Eq. (4) shows what the gradient alignment equation in Eq. (2) looks like when tuning with in-context data. For simplicity, we only show the correlation between representations which is the key in Eq.(2).
>
> * By the analysis in Section 5.2, Eq. (4) considers gradient alignment between training and unseen data $\mathcal D_{unseen}$ (which is in our objective Eq. (1)) by the attention correlation $\phi$ between samples. This is different from improving gradient alignment by FT/MTL which does not include $\phi$. We add explanations in the revision line 333-337.
>
> * We further validate the improvement of our model through the thorough experimental evaluation.
>
> **W2.2 Fig. 2 clarification**
>
> Sorry for the confusion. In Fig. 2, the correlation is between the pre-trained representation alignment (**no tuning**, using pre-trained Llama 3.1) and the model’s direct transfer performance (only fine-tuning on English data and evaluating on other languages’ data). The details are in the appendix B.1 and we also emphasize using pre-trained representations in the revision.
>
> [1]  Riemer et al.  Learning to learn without forgetting by maximizing transfer and minimizing interference. ICLR 2019.

---

### Official Review · Reviewer_z7eL · 2025-10-31

**Soundness:** 2
**Presentation:** 2
**Contribution:** 2
**Rating:** 4
**Confidence:** 3

**Summary:**

This paper introduces a framework for analyzing cross-lingual knowledge transfer in large language models (LLMs) through the lens of gradient alignment. The authors formally connect representation alignment with gradient alignment and propose a method called CL-ICP (Cross-Lingual In-Context Prompt tuning), which combines in-context learning with soft prompt tuning and representation shift to enhance cross-lingual generalization. The method is evaluated on several cross-lingual benchmarks (XQuAD, XCOPA, XNLI) and compared against fine-tuning, multi-task learning, and other methods such as ShifCon.

**Strengths:**

1. The use of soft prompts in combination with in-context demonstrations and a representation shift mechanism is an effective approach to improve alignment.
2. The experiments span diverse tasks and languages.

**Weaknesses:**

1. The central theoretical framing of this paper—gradient alignment—is insufficiently motivated. While the idea is intuitively plausible, the paper does not rigorously justify why gradient alignment should be the primary lens for understanding cross-lingual transfer.
2. The experimental setup uses a subset of available languages (e.g., only 4 languages in XQuAD/XNLI) and further restricts XNLI to just 1,000 samples without justification. These choices reduce the representativeness of the evaluation and raise concerns about the robustness and generalizability of the reported results.
3. The paper omits comparisons to several important baselines in cross-lingual transfer, such as XLM-R or other prompt-based transfer models, which makes it harder to contextualize the improvements. (XLM-R shows impressive performance on XNLI)
4. The three subsections in the related work (Cross-Lingual Transfer, Cross-Lingual Alignment Models, and Cross-Lingual ICL) have overlapping content. This structure leads to redundancy and makes it difficult to situate this paper clearly.

**Questions:**

1. Line 64 contains a likely typo (“witn” instead of “with”).
2. There is one formatting error (Lines 250–251) where the mathematical expression overflows the page width, violating the guideline.

---

> ### Author Response · Authors · 2025-11-20
>
> We thank the reviewer for suggestions about presentation and experimental settings. We address each point below.
>
> **W1 Theoretical framing**
>
> First, we would like to clarify that in the paper we provide a gradient perspective to understand the effect of alignment in cross-lingual transfer. We do not claim such a gradient perspective is the ‘primary lens’. We show the motivation of gradient alignment below.
>
> The motivation of gradient alignment comes from **the goal of the cross-lingual knowledge transfer** and **the training nature of machine learning models**.
>
> * Cross-lingual knowledge transfer aims to use a high-resource language to improve the accuracy of a low-resource language in a task [1]. The training of current ML models updates parameters by gradient descent on loss functions. Connecting these two, the gradient to reduce the loss of training data (from high-resource language or mixed language) is expected to be in the same direction as the gradient to reduce the loss of the target low-resource language’s data (i.e., improve the accuracy of the low-resource language in a task).
>
> * How the gradients are in the same direction is quantified by the inner product between their gradients. To maximize the transfer effect, it is ideal to maximize the gradient inner product above, which is our gradient alignment objective. As cited in the paper, the similar objective has been used in [2] to improve models’ cross-task transfer. We add more explanations in revision line 139-145.
>
> **W2 Experimental settings**
>
> We explain our experimental settings below. The explanation is also added in our revision line 394-396.
>
> * **Selecting subsets of languages for each task:** selecting subsets of languages in cross-lingual transfer is applied in other papers including [3, 4]. In the paper, we focus on the cross-lingual transfer to non-dominant languages and select languages with different levels of dominance (e.g., zh-high dominance; tr-dominance; sw-low dominance and qu-very low dominance) [3] and from different language systems for thorough evaluation.
>
> * **Sample 1000 data for XNLI training:** this sampling is based on our goal of cross-lingual transfer to non-dominant languages, which usually do not have high resource data in real applications. Also, we fine-tune LLM for each pair of languages on each dataset with 3 random seeds, which requires high computation cost. The same subsampling of 1000 data is also used in [5] for fine-tuning analysis and in [6] for LLM inference efficiency.
>
> **W3 Comparison to more baselines**
>
> We study the knowledge transfer in post training stage between supervised data. And we select baselines designed for this setting.
>
> **Results for prompt tuning and other baselines:** we compare with full fine-tuning (FT), prompt tuning (PT) and LoRA on XCOPA as shown below (based on Llama 3.1). We also add this in our revision Table 5.
>
> | |En-Zh|En-Tr|En-Sw|En-Qu|Avg|
> |---|---|---|---|---|---|
> |MTL (FT)|83.5|65.4 |53.1|48.4|62.6|
> |CL-ICP (FT)|88.3|77.7 |58.7|50.9|68.9|
> |MTL (PT)|56.4|55.8|52.6|50.6|53.8|
> |CL-ICP (PT)|87.6|77.9 |58.8|49.1|68.4|
> |MTL (LoRA)|87.4|75.6|59.7|50.0|68.2|
> |CL-ICP (LoRA)|88.2|81.5|63.3|53.1|71.5|
>
> * In practice, prompt tuning only may be hard to train. We show results of using prompt tuning without in-context samples, and the model turned out to have low capacity in solving the tasks. The results in the table are the best from the settings in Appendix B.2 line 971-980.
> * LoRA improves the model performance compared to full FT. However, training with in-context samples still achieves better performance than MTL with LoRA. This validates the efficacy of our CL-ICP model.
>
> **Comparison to XLM-R:** we do not think it is a parallel comparison between our CL-ICP to XLM-R itself. Our CL-ICP model is a post-training model that studies how to transfer knowledge on the supervised downstream tasks (i.e., same task in different languages). However, XLM-R uses an unsupervised method to continually pretrain the model for better direct transfer capacity. **The ways to improve the model’s pre-trained cross-lingual capacity is not the main goal of this paper**.
>
> * Since we discuss the alignment and its effect on direct transfer in Section 3, we compare XLM-R-base’s direct transfer performance and its representation alignment ability (cosine deviation score) on XNLI, shown below. Better direct transfer performance consistently accompanies lower cosine deviation, which supports our conclusion in Section 3.
>
> | Language|Cosine Dev|Accuracy|
> |---|---|---|
> |El|0.0019|77.5|
> |Zh|0.0028|76.7|
> |Tr|0.0031|74.2|
> |Sw|0.0034|66.5|
>
> * For fine-tuning analysis, we admit it’s interesting to use XLM-R as a backbone like Llama 3.1. However, XLM-R is not a generative model and requires different input-output data formats. So it may not be a fair comparison between XLM-R to our generative LLM based models.

---

> > ### Author Response · Authors · 2025-11-20
> >
> > **W4 Writing issue**
> >
> > Thank you for the writing suggestion. We reorganize the related work part (Section 2) in the paper to better show our contribution; we also fix typos and exceeding equations in the revision.
> >
> > &nbsp;
> >
> > [1] Lin et al. Choosing Transfer Languages for Cross-Lingual Learning. ACL 2019.
> >
> > [2] Riemer et al.  Learning to learn without forgetting by maximizing transfer and minimizing interference. ICLR 2019.
> >
> > [3]  Li et al. Improving In-context Learning of Multilingual Generative Language Models with Cross-lingual Alignment. NAACL 2024.
> >
> > [4] Zhang et al. ShifCon: Enhancing Non-Dominant Language Capabilities with a Shift-based Multilingual Contrastive Framework. ACL 2025.
> >
> > [5] Tu et al. Prompt-Tuning Can Be Much Better Than Fine-Tuning on Cross-lingual Understanding With Multilingual Language Models. EMNLP 2022.
> >
> > [6] Zhao et al. AdaMergeX: Cross-Lingual Transfer with Large Language Models via Adaptive Adapter Merging. NAACL 2025.

---

### Official Review · Reviewer_21p8 · 2025-10-31

**Soundness:** 3
**Presentation:** 2
**Contribution:** 3
**Rating:** 4
**Confidence:** 4

**Summary:**

This paper investigates the problem of cross-lingual knowledge transfer in LLMs, particularly focusing on how to improve task performance in non-dominant languages using knowledge learned from dominant languages . The authors frame this as a gradient alignment problem, showing that successful transfer depends on the alignment of gradients and representations between languages. They propose a novel method called Cross-Lingual In-Context Prompt Tuning (CL-ICP), which uses soft prompts and representation shifts to improve alignment, especially in low-resource settings. The method is evaluated on three multilingual datasets and compared against various baselines.

**Strengths:**

1. I think the introduction of the gradient alignment perspective to cross-lingual transfer provides a fresh and insightful view.
It connects gradient alignment with representation alignment, offering a unifying view of why some models transfer better than others.
2.The experiments are thorough, covering multiple tasks, languages, and model architectures (Llama 3.1, Qwen 2.5). The authors also include ablations and low-resource scenarios, which adds credibility to the claims.

**Weaknesses:**

1. I feel the innovative of this paper is limited:  the proposed CL-ICP combines existing ideas (in-context learning, soft prompts, representation shifting) rather than introducing fundamentally new mechanisms.
2. CL-ICP relies on demonstration selection and attention analysis, which may not scale well to more complex tasks or longer sequences. The paper does not discuss computational overhead or inference efficiency, which are critical for real-world deployment.
3. The overall display of the results could be improved, most of the tables are too small to see, and some equation exceeds the width limit.

**Questions:**

see weakness

---

> ### Author Response · Authors · 2025-11-20
>
> We thank the reviewer for recognizing gradient alignment is a fresh and insightful view to cross-lingual transfer and suggesting the improvement of analysis and presentation. We address each point below.
>
> **W1 Novelty of the paper**
>
>  The novelty of our paper includes but is not limited to the CL-ICP model structure.
>
> * We consider our main novelty is to provide a new and unified perspective to study the alignment in cross-lingual knowledge transfer, as you mentioned in the strength section. In comparison, previous works do not study the alignment effect of existing training strategy and mainly focus on the representation alignment. **However, why and how representation alignment is necessary for cross-lingual transfer are underexplored; and it is unclear why simple training strategies can already perform well in many cases**. Our work addresses these important questions.
>
> * Based on the perspective, CL-ICP is designed to improve cross-lingual alignment in both gradient and representations. We show the distinction between CL-ICP and previous training strategies (FT and MTL) in addressing the gradient alignment in Section 5; and provide a novel way to add representation shifts in the in-context training. **To the best of our knowledge, the gradient alignment effect and the representation shift in the in-context training has not been studied before. Moreover, the improved performance of CL-ICP validates our novel perspective of alignment in the paper**.
>
> **W2 Computational overhead**
>
> * **Cost of analysis**: we conduct the attention and demonstration analysis to understand the mechanics of in-context cross-lingual alignment, which we believe is also a contribution of the paper. The analysis is conducted by in-context learning on the pre-trained LLM, which requires no parameter updating and only needs inference computation (discussed below).  **After the analysis, we adopt a unified training strategy for different tasks and languages without extra analysis cost**.
>
> * **Cost of training and inference**: the computation overhead mainly comes from the growing sequence length after adding the in-context demonstrations. In the work, we already reduce the computation by the following:
>
>   * For complex tasks/long sequences like XQuAD, using less/truncated demonstration samples.
>
>   * Learning soft prompts instead of full-fine-tuning. Without sacrificing much of the performance, using prompt tuning can still achieve good performance but reduce the computation cost.
>
>   * At the inference time, using KV cache to store the states of demonstrations and prompts, which saves computation at each step’s token generation.
>
> &nbsp;&nbsp;&nbsp;&nbsp; The additional computation cost during training and inference time is a common problem for in-context models. Since in-context models are widely used in NLP applications, we believe future works will make consistent efforts to better address the computation cost issue.
>
> **W3 Writing improvement**
>
> Thank you for the suggestion about writing. We have shortened the equations and increased the size of Table 2-4 in the revision.

---

### Note · Authors · 2026-01-06

I have read and agree with the venue's withdrawal policy on behalf of myself and my co-authors.